# Virus Pop—Expanding Viral Databases by Protein Sequence Simulation

**DOI:** 10.3390/v15061227

**Published:** 2023-05-24

**Authors:** Julia Kende, Massimiliano Bonomi, Sarah Temmam, Béatrice Regnault, Philippe Pérot, Marc Eloit, Thomas Bigot

**Affiliations:** 1Bioinformatics and Biostatistics Hub, Institut Pasteur, Université Paris Cité, F-75015 Paris, France; 2Department of Structural Biology and Chemistry, Institut Pasteur, Université Paris Cité, CNRS UMR 3528, F-75015 Paris, France; 3Pathogen Discovery Laboratory, Institut Pasteur, Université Paris Cité, F-75015 Paris, France; 4Ecole Nationale Vétérinaire d’Alfort, F-94700 Maisons-Alfort, France

**Keywords:** sequence simulation, sequence evolution, phylogenomics, amino acid substitution rates, database

## Abstract

The improvement of our knowledge of the virosphere, which includes unknown viruses, is a key area in virology. Metagenomics tools, which perform taxonomic assignation from high throughput sequencing datasets, are generally evaluated with datasets derived from biological samples or in silico spiked samples containing known viral sequences present in public databases, resulting in the inability to evaluate the capacity of these tools to detect novel or distant viruses. Simulating realistic evolutionary directions is therefore key to benchmark and improve these tools. Additionally, expanding current databases with realistic simulated sequences can improve the capacity of alignment-based searching strategies for finding distant viruses, which could lead to a better characterization of the “dark matter” of metagenomics data. Here, we present Virus Pop, a novel pipeline for simulating realistic protein sequences and adding new branches to a protein phylogenetic tree. The tool generates simulated sequences with substitution rate variations that are dependent on protein domains and inferred from the input dataset, allowing for a realistic representation of protein evolution. The pipeline also infers ancestral sequences corresponding to multiple internal nodes of the input data phylogenetic tree, enabling new sequences to be inserted at various points of interest in the group studied. We demonstrated that Virus Pop produces simulated sequences that closely match the structural and functional characteristics of real protein sequences, taking as an example the spike protein of *sarbecoviruses*. Virus Pop also succeeded at creating sequences that resemble real sequences not included in the databases, which facilitated the identification of a novel pathogenic human circovirus not included in the input database. In conclusion, Virus Pop is helpful for challenging taxonomic assignation tools and could help improve databases to better detect distant viruses.

## 1. Introduction

In virology, metagenomics tools performing taxonomic assignation from high throughput sequencing datasets are currently key for detecting unknown viruses and improving our knowledge of the virosphere (e.g., [1,2,3]). They are generally evaluated with datasets derived from biological samples or in silico spiked samples, which all contain known viral sequences present in public databases. This considerably limits the evaluation of their capability of finding distant viruses. Given the high degree of variability in viral protein sequences, simulating realistic evolutionary directions appears to be key to challenge tools in their capability to identify such distant viruses. Going further, expanding the current databases by realistic simulated sequences could improve the capacity of alignment-based searching strategies for finding distant viruses.

The simulation of realistic biological sequences has been instrumental in bioinformatics for some time. For instance, phylogenetic method evaluation [4,5], machine-learning tool training [6] and, when data are limited, hypothesis testing [7] have been relying on simulated data. Following the development of advanced methods for inferring substitution matrices and complex substitution models, several protein sequence simulation tools have been developed. Generally, simulations will mimic evolutionary processes with random draws, deciding for amino-acid mutations over time (Appendix B). The random draws follow probability laws that reflect the observed chances of mutation from one amino acid to another, but they do not take into account that biological constraints can occur at specific positions, which limit the spectrum of possible mutations. Some models can also simulate insertions and deletions.

Two of the most comprehensive and commonly used programs are Seq-gen [8] and IQTREE [9,10]. Both allow the user to choose a substitution model or to input one.

However, the currently available tools generate data mimicking evolutionary parameters in a statistical way, without maintaining site specificities. For instance, if a sequence alignment is composed 50% of quickly evolving sites and 50% of invariant sites, the simulation will respect the 50/50 ratio but will randomly decide which site evolves quickly and which does not. As a result, although simulated sequences can be considered as representative of a clade in terms of evolution parameters, they could not be considered as homologous to real proteins. Furthermore, the currently available tools usually offer the possibility to fill a given tree with simulations, starting from one input sequence as the tree root. However, no tool provides a practical way to populate a tree with new branches at various positions.

Here, we introduce Virus Pop, a pipeline for adding new branches to a protein phylogenetic tree. Virus Pop leverages the tools provided by the last version of IQ-TREE [9] and integrates them in a comprehensive process that simulates protein evolution in a realistic way in terms of site-dependent substitution rate in the input dataset. From collecting the input dataset of existing proteins to retro-translating simulated sequences, several options are available to simplify the construction of the starting dataset, and configurable steps allow the user to precisely constrain the locations of the generated sequences within the tree. The outcome is a set of simulated sequences that complete the initial tree with new branches connected at the desired topological position. The evolutionary distances simulated, represented by the new branch lengths, can either be set automatically based on observed distances in the starting dataset, or manually specified.

We evaluated the realism of the simulated sequences by examining the conservation of protein structure in generated sequences of the *Sarbecovirus* spike protein. We also illustrated two examples of how simulated sequences with varying parameters and increasing distances can serve as input data to evaluate Blast-based assignation. Finally, we showed that simulated sequences may be close to real sequences not yet indexed in databases and that the addition of such sequences in databases used for taxonomic assignment could help to reduce the number of unassigned reads in metagenomic data.

## 2. Materials and Methods

### 2.1. The Virus Pop Pipeline

Virus Pop was developed as a simple command-line program with Python and Snakemake [11]. The pipeline uses the Dendropy [12], Ete3 [13], Biopython [14] and iTOL [15] python libraries. The main steps of the pipeline are as follows:Building the starting dataset. Many options are provided to offer convenient ways for the user to quickly build a dataset. At the end of this step, Virus Pop will have one or multiple fasta files, each representing a protein type and containing homologous sequences;Finding the best evolutionary model and phylogenetic tree for each set of homologous proteins;Choosing nodes (automatically or manually) in the phylogenetic tree and reconstructing ancestral sequences at these node positions;Simulating the evolution of the reconstructed ancestral sequences (retro-translated nucleotide sequences are also provided).

Figure 1 illustrates the full pipeline. Each step is detailed hereafter.

#### 2.1.1. Building the Starting Homologous Protein Dataset

The user can directly input protein sequence fasta files, however, Virus Pop provides an automated dataset construction tool for helping the user to gather sets of aligned homologous protein sequences. Each set must be large enough to be representative of the variability amongst the group of interest, while remaining small enough to allow reasonable time-efficiency. This tool constitutes a quick and easy solution for working on any virus group and protein type, without the need to manually select the sequences.

The input parameter for automatic dataset construction is the name of a taxonomic group. The process can be decomposed in the following steps:Parsing the NCBI taxonomy tree to extract all species belonging to the taxonomic group entered;Requesting, from NCBI, the number and identifier of available complete genomes in each species and selecting randomly a list of identifiers within each species;Fetching the selected genomes;Clustering proteins: running BLASTp [16], then extracting from the results clusters of homologous protein sequences with SILIX [17]. SILIX is run with the following parameters: 20% minimum identity and 60% minimum overlap; and BLAST is run with a cost of 1 and 9 to open and close gaps, respectively;Aligning the sequences in each cluster with MAFFT [18] (with default parameters);Generating a descriptive protein name for each cluster, based on word frequency in the protein annotations.

The result of the fetching tool is a ready-to-use dataset of aligned homologous protein clusters from the virus taxonomic group of interest with descriptive names that will help the user to know which proteins were extracted.

Alternatively, the user can provide a list of taxa to download, a set of full genomes (complete nucleotide genomes and protein sequences in fasta files), or fasta files with homologous proteins. The data preparation process will then start from steps 3, 4, respectively, or it will have nothing to do.

#### 2.1.2. Finding the Best Evolutionary Model

The first step of the evolutionary model construction is to infer a substitution model with site-dependent rates. This process also generates a phylogenetic tree. Virus Pop uses the Model Finder tool provided within IQTREE [19]. Except for the constraints on the type of substitution model wanted, all other parameters are left to their default value. The construction of the model is as follows:The Model Finder tool is used in basic mode (no rate variation and no invariant site), to find in a limited amount of time the substitution model best representing the alignment. Let *M* be the best model found.The Model Finder tool is used again, but constrained so that it will fit a 5-discretized gamma profile with the *M* model: M+5G. A 5-discretized gamma profile is an evolutionary model that differentiates five substitution rate categories. Each locus in the protein considered is associated to one category. This is outputted in the form of a file which indicates, for each locus, the mean substitution rate corresponding to the category to which it belongs. Although the optimal number of bins is dependent on the dataset, we used five as it yields good results in most cases [20].The mean substitution rate distribution is retrieved and turned into a partition description file. Originally, partition files were meant to define different segments along an alignment that will follow different substitution models. Here, we use this option but only the substitution rate varies. Tweaking with partition models is a way to simulate site-dependent substitution rates without fully implementing a new method.Using Model Finder one last time, constrained with the partition file, a phylogenetic tree is inferred with the ancestral sequence statistics. This consists of a file with, for each internal tree node and each site, the probabilistic amino acid distribution.

At this point, each set of homologous protein is described with a substitution model, a site-dependent substitution rate profile, a phylogenetic tree and the probabilistic composition of ancestral sequences.

### 2.2. Choosing Internal Node of Interest and Building Ancestral Sequences

The next step is to choose the nodes at which ancestral sequences will be reconstructed. The chosen node may vary depending on the purpose of the user. In most cases, however, it is of no interest to simulate within a subgroup of very closely related sequences. Thus, nodes of interest are usually internal nodes that are the common ancestor of, at least, a whole clade of related sequences. By whole, we mean that no other sequence is very close to the group.

Virus Pop provides a way to automatically select nodes according to this strategy. Based on a maximum distance, it creates clusters of sequences in which one sequence cannot be more distant than a defined maximum distance to the closest sequence within the group. It then selects the node representing the most common ancestor of each group. Figure 2 illustrates this operation with a varying distance threshold.

This step is a bottleneck in the pipeline at which it is recommended that the user checks the nodes selected automatically. The user should visualize the phylogenetic tree and, if needed, manually select the desired nodes. The Virus Pop project includes a tool for automatically loading the constructed trees in iTOL [15] and facilitate this step. Alternatively, if the purpose of the user is to create a great diversity of new sequences, they may want to populate the tree with new branches at every single internal node. Virus Pop also provides this option.

Finally, with the selected nodes and inferred statistics of ancestral nodes provided at the previous step (see Section 2.1.2), Virus Pop reconstructs ancestral sequences: the user can choose to build single majority-rule consensus sequences for each selected node. In this case, Virus Pop will select the most probable amino acid at each site. Alternatively, Virus Pop randomly picks amino acids based on the statistics for each site. In the first case, only one sequence reconstruction is built. In the second case, any number of ancestral sequences can be constructed.

#### 2.2.1. Evolutionary Distance, Simulation and Gaps

With the partition model and ancestral sequences, the last requirement for performing the simulations is to choose the evolutionary distances that will be simulated. Virus Pop provides two ways to define these distances: the first one is to manually input distances as a list; otherwise, Virus Pop will automatically generate distances. Considering one ancestral node and N requested simulations from each ancestral sequence, Virus Pop will simulate distances regularly distributed within half and twice the mean distances from the ancestral node to its child leaves.

Simulations are then performed with one of IQTREE tool: ALISIM [10]. This tool can take a root sequence, a partition model and a tree, and will populate the tree leaves, respecting the model, the topology and the distances. However, in our case, the objective is to constrain the branching of new sequences within the topology of the real data phylogenetic tree. The position of the simulated sequence in the subtree is left to the randomness of evolution simulation. To achieve this, simulations are performed on “tiny trees” sequentially constructed as presented in Figure 3.

Each “tiny tree” is a simple tree-node graph, with an ancestral sequence at the root and two empty leaves on branches with the desired simulation distance. The position within the complete tree is constrained thanks to the ancestral sequence construction. Running ALISIM will generate two sequences (S1 and S2 in Figure 3), but only one (selected randomly) is kept and integrated to the pipeline outputs.

Because this method generates sequences with amino acids on all sites of the alignment, we implemented a method to guaranty that a region that is always a gap in some part of the phylogenetic tree will not be filled randomly. This post-processing adds gaps presenting the subtree in which the simulation is occurring. Based on each gap frequency in this subtree, a weighted random draw is performed to decide whether to reproduce the gap or not in the simulation. However, no insertion or deletion pattern that was not observed in the subtree is simulated.

#### 2.2.2. Retro-Translation

Because one of the main purposes of the development of Virus Pop is to provide datasets for characterizing, testing and improving taxonomic tools that often operate on metagenomic sequencing reads, the user may need nucleotide sequences. Virus Pop thus provides a retro-translation of the simulated sequences.

The retro-translation is based on a probabilistic analysis of the codon usage in each amino-acid neighborhood, within the alignment. The principle is as follows:
**for** each site *i* with amino acid *AA*
k←0**while** no corresponding codon usage is found
look for coding of AA in sites [i−k:i+k]k←k+1**draw** a codon amongst found codons

This simple approach will provide retro-translations that preserve possible variations of the codon usage along the sequences. Indeed, if an amino acid at one site is always coded with the same codon, the retro-translation method proposed will maintain this characteristic.

### 2.3. Datasets

We used two datasets for presenting and testing the pipeline.

#### 2.3.1. *Sarbecovirus* Spike Protein Dataset

The *Sarbecovirus* dataset was hand-built to be representative of the *Sarbecovirus* sub-genus variability, with a special focus on the SARS-CoV-2 clade, including bat-related *sarbecoviruses*. It consists of 300 complete genomes, from two different sources: GISAID [21] and GenBank [22].

We translated the viral genes to protein sequences and selected the spike protein sequences from each genome. As the spike protein includes a well-characterized receptor binding domain (RBD), which binds to the human ACE2 protein, it is an adequate protein for testing the simulation process: first, because the structure of the RBD-ACE2 complex is well-characterized at the amino-acid level since the emergence of SARS-CoV-2, and mainly because the study of its evolution and variation is a key example that illustrates the comprehension of human susceptibility to viruses from animal reservoirs.

Figure 4A presents the phylogenetic tree constructed from the spike amino acid sequences with IQTREE [9]. The SARS-CoV-2-like group is outlined.

#### 2.3.2. The *Circovirus* Capsid Protein Dataset

To test Virus Pop in the context of a protein sequence more variable than the spike protein of *sarbecoviruses*, we worked on the capsid protein of the members of the *Circovirus* genus (*Circoviridae* family) which are known to infect a large spectrum of animals, including a first case of human infection we recently discovered [23]. The dataset was built with the automatic dataset construction tool. As we later tested the possibility to predict recently discovered *Circovirus* sequences based on this dataset, we manually checked that the database did not include the new genomes [23,24].

As the NCBI database includes a high number of species from the *Circovirus* genus, we used the –target_n_genome option to limit to 1 the number of genomes fetched within each species. The –min_cluster_size option was set to 30 to limit the substitution model inferences to protein clusters containing at least 30 sequences.

The complete command is as follows:


./run_virus_pop.py group_name Circovirus Circovirus_project -t 1 -s 30


The result is a dataset of 97 *Circovirus* complete genomes with annotated proteins. The clustering detects 4 homologous groups amongst which the two biggest are the replication (86 sequences) and the capsid protein (65 sequences). Based on the annotations, the automatic name construction respectively described them as “replication [associated] protein” and “capsid protein”. The dataset is available in Appendix A.

Figure 5 shows the phylogenetic tree constructed from the capsid *Circovirus* sequences.

### 2.4. Performances

The pipeline was developed and tested on an Ubuntu system with 32 GiB RAM and Intel Core i9 with 16 cores at 2.40 GHz. Most steps within the pipeline exploit the availability of multiple cores. Our tests have shown that more than 8 cores are rarely needed, so this number was set as the maximum core number allocated to one job. Furthermore, thanks to the Snakemake framework, if more than one homologous protein is identified and if enough cores are available, Virus Pop will automatically parallelize the computation of multiple protein simulations.

When launched for the simulation of the *Sarbecovirus* spike protein sequences presented in Figure 4B, it ran in 53 min. Most of this time is related to building the evolutionary model. Reconstructing ancestral sequences and simulation only take up 3 s of the total time. This means that once the model is inferred, the user can quickly test different simulation parameters. For the *Circovirus* dataset presented in Figure 5B, it ran in 22 min with 1.5 min to build the dataset and only 8 s to generate the simulations.

However, when required to simulate a high number of nodes, the time needed for the simulation part increases. For instance, to generate 100 ancestral sequences at each selected node presented in Figure 4A and simulate 3 evolution distances from each reconstructed ancestral sequence, it takes 28 s.

### 2.5. Structure of the Simulated Spike Protein Receptor Binding Domain

To assess the biological significance of the simulated protein sequences, we quantified in silico the affinity of the RBDs of the simulated spike sequences to the human ACE2 (hACE2) receptor using the following pipeline.

We used AlphaFold2 [25] to build structural models of the hACE2 and of the spike RBD of each simulated sequence.We created models of the hACE2-RBD complex using a local installation of HADDOCK [26]. hACE2 and RBD were first docked into a complex guided by inter-subunits distance restraints extracted from the X-ray structure of the SARS-CoV-2 RBD bound to hACE2 (PDB code 6M0J; [27]). The models were then refined by short MD simulations in explicit solvent during the last step of the HADDOCK modelling pipeline. For each RBD sequence, we created 200 models of the hACE2-RBD complex.The empirical scoring function FoldX v. 5 [28] was used to estimate the RBD-hACE2 binding free energy.

## 3. Results

### 3.1. Phylogeny of Simulated Sequences

To verify that the constraints on the topology and evolutionary distances of simulated sequences were respected, we compared the phylogenetic tree before and after adding the simulated sequences. We performed this test on the *Sarbecovirus* spike (Figure 4) and *Circovirus* capsid protein datasets (Figure 5). The complete sequences and Newick trees are available in Appendix A.

For the *Sarbecovirus* spike protein, ten ancestral nodes were selected for sequence reconstruction. Then, evolution was simulated on each ancestral sequence with default parameters for the number and distances of simulation. As a result, three sequences were generated from each ancestral node with a distance varying from 0.5 to 2 times the median distance in the ancestral node subtree. This automatic setting generates distances comparable to the ones observed in the tree (Figure 4). Furthermore, all simulated sequences are branched at the desired position.

In the case of the *Circovirus* capsid protein, we selected nine ancestral nodes and generated fixed distances of 0.1, 0.5, 1, 2, 3, 4 and 5. The resulting trees are presented in Figure 5. For this test viral genus, we imposed distances far from the observed distances. In all but one case, the simulations were branched below their corresponding ancestral node.

Nevertheless, in most cases Virus Pop succeeds at populating the phylogenetic trees with new sequences at the desired position and distances. This provides an additional argument to verify the realism of the simulated sequences, showing that they represent realistic possibilities in terms of an evolutionary scenario.

### 3.2. Biological Significance of Simulations: Structural Evaluation of the *Sarbecovirus* Spike Dataset

We completed the phylogeny visual inspection presented in Section 3.1 with an evaluation of the affinity of the simulated spike protein receptor binding domain (RBD) for the human ACE2 (hACE2) receptor. For this calculation, we used a computational pipeline that combines structure prediction from sequence with AlphaFold2 [25], assembly of the hACE2-RBD complex with HADDOCK, and binding free-energy estimation with FoldX (see Section 2.5). We consider that a well-predicted affinity is an indicator that the structure of the RBD is preserved enough for binding to hACE2 to be possible. This would mean that the simulated evolution went in a realistic direction and can then be under positive selection pressure but does not inform on the conservation of the whole spike protein structure, which can also influence the binding of the RBD. The result of the pipeline is presented in Figure 6.

We present the binding free energy of the complex as a function of the RBD sequence identity. Full data are available in Appendix A as well as a visualization of five predicted structures (Appendix A). The first striking result is that there is no correlation between the sequence identity and the ability of simulated sequences to bind the human ACE2 receptor, as represented by the binding free energy (correlation coefficient of 0.08). This indicates that the evolutionary model constructed is accurate enough to allow simulating long evolutionary distances while preserving the physicochemical properties of the RBD domain. Moreover, amongst the 240 tested sequences, 61 ( 25%) have a dissociation constant (Kd) below 1 µM (equivalent to a binding energy below −8.175 kcal/mol at 298 K) which corresponds to a stable complex. Almost 10% of the sequences have an estimated binding energy even greater than the reference SARS-CoV-2 RBD, meaning that they could bind hACE2 with a better affinity than SARS-CoV-2, as in the case of several bat coronaviruses close to SARS-CoV-2 [29]. These results demonstrate that Virus Pop is efficient in producing viral sequences that could be under positive selection pressure and potentially found in nature.

### 3.3. Virus Pop for Tools and Pipeline Testing

To illustrate how Virus Pop can be used for challenging and comparing taxonomic assignation tools, we tested the alignment results of BLASTp [16] and DIAMOND [30] with our simulated sequences against our initial set of real sequences. The purpose was to illustrate how the local alignment performance varies with simulated distances and how different tools yield different performances. We chose BLASTp and DIAMOND as they are widely used for local sequence alignment of protein sequences and are often integrated within taxonomic assignation pipelines (e.g., [1,2,3]). Depending on the software used within each taxonomic assignation pipeline, their parameters and cut-off values, their performances will be affected in a different way.

The results presented here were obtained on our two datasets. We generated 100 ancestral sequences at each selected internal node position (see Figure 4 and Figure 5) and simulated evolutionary distances of 0.1, 0.5, 1, 2, 3, 4 and 5 from each reconstructed ancestral sequence. Testing the generated sequences with BLASTp and DIAMOND, we kept 25 hits per query but discarded the ones with a bit score below 40 [31]. Diamond was launched with its “sensitive” mode. For the sake of clarity, we depict here the complete simulated sequences, corresponding to a case in which NGS read assembly creates contigs corresponding to the full proteins. Figure 7 and Figure 8 present the ratio of aligned sequences along the alignment, as well as the size of the alignments returned by BLAST and DIAMOND.

As presented in Figure 7, on the highly variable *Circovirus* capsid protein, BLAST and DIAMOND perform similarly up to a distance of 1 on our test. Beyond that, the alignment coverage drops quickly, with DIAMOND performing worse than BLAST. On the *Sarbecovirus* spike protein, because many sites are almost invariant, the performances of the two tools are similar. The only slight alignment number decrease is visible before and at the very beginning of the NTD region (red arrow in Figure 8). However, this does not constitute a formal benchmarking of those tools as we present results here on a limited amount of data, and as they could both be tested with different parameters that include computing times. For instance, the “ultra sensitive” mode of DIAMOND would produce alignment results almost similar to those of BLAST. The purpose here is to illustrate how Virus Pop can help in assessing the performance of a tool or a full pipeline. Depending on the input data type, time performances or other constraints, it can participate in making development choices.

### 3.4. Improving Taxonomic Tool Sensibility by Expanding Reference Databases

Beyond testing and evaluating taxonomic assignation tools, an application of Virus Pop that we put to the test was the interest of using the simulations to expand databases. We hypothesized that if the simulations represent realistic evolution directions, then they may be closer to yet unobserved viruses than any of the indexed viruses. Thus, integrating simulations to databases may increase the range of the detection tool.

We tested this hypothesis on two recently discovered *Circovirus* capsid proteins [23,24] that were not part of our initial dataset. To mimic in a simple and deterministic way the read alignment issue, we generated all 50-amino-acid-long segments of the two proteins. We obtained a total of 330 fake “perfect” reads (165 for each of the two 214 amino-acid long proteins).

For the simulation part, as we were in a context where we wanted to expand the tree at all possible positions and in many evolutionary directions, we generated 200 ancestral sequences for each of the 130 internal nodes. We used the default parameters for the distances which means that three sequences were created from each ancestral sequence. As a result, we obtained 600 simulations under each internal node which corresponds to a total of 78,000 simulated sequences. The sequences are available in Appendix A

Finally, we compared the result of blasting the protein segments against the initial database and against the augmented database. We discarded all alignments with a bit score below 40. Note that, in this protocol, the initial dataset was generated randomly with each species of the taxonomic group on NCBI being represented by only one sequence. Other indexed sequences that were not fetch in this run may be closer to the two recently discovered *Circovirus*.

Out of the 330 simulated reads, 170 were aligned at least once against the initial real database. This figure goes up to 202 against the augmented database, meaning that we aligned 10% more reads. In addition, with 6000 and 60,000 simulations at each internal node, we increase this score to 16% and 33% more reads detected, respectively.

To complete this approach and visualize how close to the targets the simulated sequences are in the scenario with 600 simulations at each internal node, we present a zoom of the augmented tree (Figure 9). It contains the two closest real sequences in the starting dataset, the 600 simulations created from their common ancestor (this internal node is highlighted in Figure 5) and the two new protein sequences. The sequences on which were aligned most of the fake reads are indicated. This verifies that Virus Pop does generate enough variability such that we create sequences close to real proteins that are not yet recorded in the database.

### 3.5. Database

We ran Virus Pop on 80 virus genera infecting humans and 110 genera affecting other vertebrates. These virus lists were obtained from ViralZone [32]. In the first group, 57 genera had a sufficient number of full genome sequences available with protein clusters detected by Virus Pop. In the second group, only 36 genera met these two conditions, allowing the pipeline to run through to completion. With the increasing availability of NGS sequencing data, simulating additional genera with Virus Pop will be possible.

For each genus, up to 15 full genomes of each species within were fetched. Then, for each homologous protein group found, 300 simulations (100 ancestral sequences times 3 automatically computed distances) were generated at each internal node of the inferred phylogenetic tree. To gather enough sequences within 3 genera with few species (*Sapovirus*, *Parechovirus* and *Hepatovirus*), the number of fetched genomes was increased to 25.

The result is a total of 995 simulated homologous protein groups with a grand total of 24,138,277 sequences in both amino acid and nucleotide. The number of homologous protein groups found and simulated within one genus ranges from 1 (for 21 genera) to 138 for *Orthopoxvirus*.

The resulting database is available at https://doi.org/10.5281/zenodo.7712690.

## 4. Discussion

Although several tools for creating fake amino-acid and nucleotide sequences have been proposed [8,10,33], none of them provide a ready-to-use method for populating a given taxonomic group with new realistic sequences. How Virus Pop tackles this shortfall is three-fold:Generating simulations with substitution rate variations depending on sites and inferred from the input dataset. Existing tools do allow to simulate substitution rate variations. For instance, IQ-TREE and Seq-gen both give the possibility to generate rate heterogeneity based on a gamma-model [34]. However, they are implemented without the possibility for the user to constrain which site will be in which gamma-rate category. As a result, for instance, if 20% of the amino acid sites evolve slowly, simulations will statistically reproduce this parameter. However, the 20% of slowly evolving sites will be randomly spread over the alignment and conserved segments may be lost. Other methods for rate heterogeneity simulations exist but, so far, no tool provides an easy way to analyze the rate heterogeneity in real dataset and to reproduce it.Inferring ancestral sequences corresponding to multiple internal nodes of the input data phylogenetic tree. This allows Virus Pop to generate new sequences that will be branched at various points of interest in the group studied, and not only new sub-tree deriving from a leaf sequence.Adapting the simulated evolutionary distances so that the new sequences will be inserted within the input phylogenetic tree at realistic distances compared to surrounding real sequences. Alternatively, the user may manually choose any distance.

Embedded in a complete pipeline with as many automatic steps as possible and multiple configurable options, Virus Pop provides a complete solution to efficiently generate new sequences within a given taxonomic group.

The visualization of the final phylogenetic tree built with both the starting set of sequences and the phylogenetic sequences shows that the simulations are inserted within the tree at the chosen positions and distances. Moreover, the use of a carefully selected substitution model generates mutations that have more chance to preserve the physicochemical properties of the protein, while site-dependent substitution rates promote the positioning of mutations in variable regions and preserves conserved regions. During our test, only long simulated branches sometimes end up branching more deeply in the tree (see the most distant yellow sequence in Figure 5). This is likely due to the difficulty of accurately positioning very long branches [35].

When checking the validity of simulated biological sequences, one must be careful to avoid redundant reasoning. Indeed, our perception of evolutionary processes is based on the same mathematical models on which bioinformatics tools are built. Additionally, simulated data, by definition, respect the mathematical models with which they are constructed. Thus, there is a risk in considering our simulated sequences as validated simply on the basis that they rely on the mathematical structure of a substitution model.

To escape this issue and test further the “realness” of the sequences, we performed a structural analysis to evaluate the conservation of a protein function: the affinity of the *Sarbecovirus* RBDs of the simulated spike sequences to the human ACE2 (hACE2) receptor. Although limited by prior functional knowledge, it constitutes a way to escape the redundancy of a reasoning based on creating and evaluating sequences with the same mathematical model. Our result shows that some sequences lost their function, but that Virus Pop is still able to generate 25% of sequences that may be functional even amongst the most evolutionary distant. This means that the precise substitution model inferred with substitution rate variations does create sequences with a predicted good preservation of their physicochemical and functional properties. Of note, experimental confirmation of this result is needed, for example by measuring the real binding affinity of simulated sequences with the hACE2 receptor.

Another solution to improve the preservation of the protein structure would be to apply a mathematical model that would integrate interactions between residues. Indeed, overall structure is partly the result of internal interactions within monomers or between multimers, with dependencies between sites that will co-evolve to preserve their interaction. Integrating the interactions within the mathematical model would amount to simulate evolution in a “structure-aware” way. However, this would mean going beyond the initial assumption of site independence on which the evolution models are based. But this drastic simplification is currently necessary as evolutionary models are inherently very complex mathematical objects. For instance, determining an empirical amino-acid substitution model, as those used in Virus Pop, consists in finding 210 parameters (see Section A.1). Having a dataset in which each type of substitution is represented in a quantity sufficient to derive representative rates is already a challenge. Thus, even if we could define models with dependencies amongst sites and if we had the computational power to solve them, we would be limited by the availability of data. There have been, however, some published efforts put into trying to incorporate protein structure constraints to account for site dependencies. However, they were defined on case-by-case observations of some amino acid interactions [36,37].

To illustrate the usefulness of Virus Pop, we then presented how precisely constrained new sequences constitute a well-defined dataset convenient for testing and comparing the performances and limits of taxonomic assignation tools. Our two examples (see Section 3.3) show how different the results are, depending on the ratio of highly preserved loci. On the *Sarbecovirus* spike protein, simulated substitutions are stacked on a few loci that, compared to the preserved areas, have a substitution rate that is more than 1000 times faster. However, even on proteins with these characteristics, a test performed with reads simulated on poorly preserved areas will tend to show decreasing performances as the distances increase. To conclude on this matter, tests should be performed on varying protein types and with varying distances to constitute a good prediction on our ability to detect novel viruses. One downfall of the Virus Pop pipeline is however that, as it is based on protein evolution models, the retro-translation step adds silent mutations that are not well accounted for in any defined parameters. Thus, Virus Pop is more suitable for assessing pipelines within which reads and contigs are translated and alignment steps are performed on amino-acid sequences, such as Microseek, which was recently released [2].

In addition, we demonstrated how generating a high number of simulated sequences (for example the *Circovirus* capsid protein) succeed at creating new sequences close enough to retrospectively identified real sequences, illustrating the usefulness of Virus Pop in discovering new viruses if simulated sequences are included in public databases used for virus taxonomic assignation. As suggested by the fact that the simulated sequences on which the tested real new proteins were aligned are scattered in the tree (see Figure 9), there are probably multiple generated sequences that are locally very close to the real ones and that participate in the detection of reads. Though the increase of alignment score is moderate, adding simulations to taxonomic assignation tool databases could constitute a strategy to lower the risk of missing viruses belonging to groups of particular concern. Moreover, as the simulations are built with highly constrained evolution models, they should not be responsible for much additional background noise.

Finally, this strategy could be improved with a filter based on an approach similar to the one presented for the *Sarbecovirus* spike protein in Section 3.2. The combination of Virus Pop predictions regarding virus evolution with corresponding predictions of protein-protein interactions inform on which signal should be the most relevant to be detected in metagenomics data. Filtering Virus Pop simulations with a structural analysis could thus also help to predict the existence of still unknown viruses with a spillover potential for humans. This way, we could also imagine purposely decreasing the constraints on our evolutionary models to allow bigger evolutionary jumps. However, the test presented here for the affinity of the RBD sequences was carefully hand-built and relies on the possibility to test the stability of a complex. Efficiently generalizing this scheme to any type of protein needs a good knowledge of their functional domains.

Alternatively, one could integrate a simple filter based on the elimination of sequences that do not preserve some residues known to be involved in the protein function. The implementation of such a solution would be much faster. However, crossing the results from Section 3.2 on the *Sarbecovirus* spike protein with the number of preserved residues shows that some simulations generated present a good preservation of the predicted affinity, although few contact residues are preserved (see Appendix A). This solution would thus reduce the liberty of the simulation process and may limit Virus Pop ability to discover functional evolutionary jumps.

In conclusion, Virus Pop is a valuable tool for generating new realistic biological sequences within a given taxonomic group. By selecting the best empirical substitution model and implementing site-dependent substitution rates, it creates mutations that tend to preserve the physicochemical properties of the protein and promotes the positioning of mutations in variable regions, while conserving preserved regions. The ability to insert new sequences anywhere within the phylogenetic tree and at realistic distances compared to surrounding real sequences makes it a complete solution for expanding viral databases with realistic sequences. The well-defined datasets provided by Virus Pop are useful for testing and comparing taxonomic assignation tools and could also be integrated in data banks in order to enhance the sensibility of pattern-based taxonomic tools for detecting unknown viruses in groups of particular concern.

## Figures and Tables

**Figure 1 viruses-15-01227-f001:**
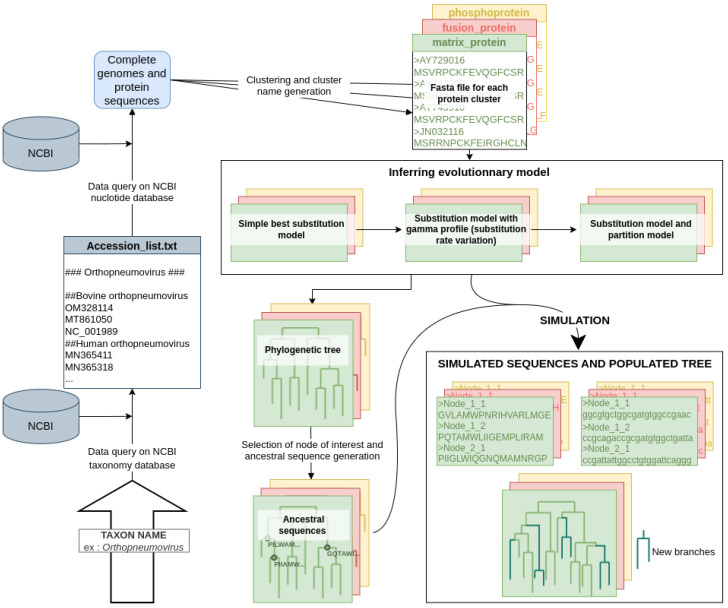
The Virus Pop pipeline. This figure presents the complete pipeline, starting with only the input of a taxonomic group taken as an example: the *Orthopneumovirus* genus. However, the user can provide a NCBI accession ID list, genome fasta files (both the complete nucleotide genome and protein sequences in fasta files) or files of homologous proteins. The final outputs are simulated sequences, in both nucleotides and amino acids, for each group of homologous proteins.

**Figure 2 viruses-15-01227-f002:**
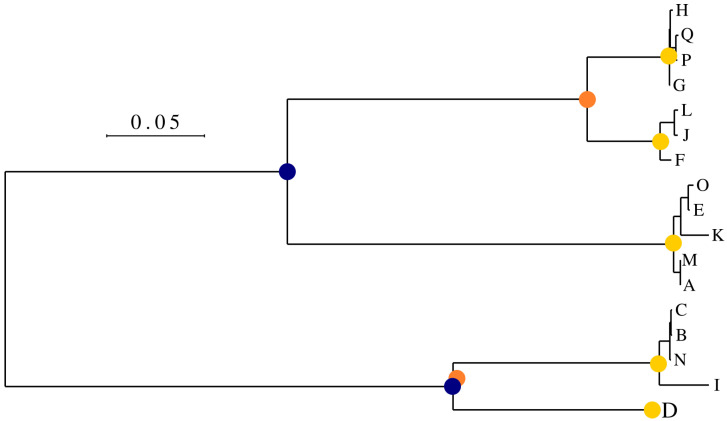
Example of automatic node selection by leaf clustering. Solid circles represent the selected nodes for a maximum distance of 0.025 (yellow), 0.2 (orange) and 0.25 (blue).

**Figure 3 viruses-15-01227-f003:**
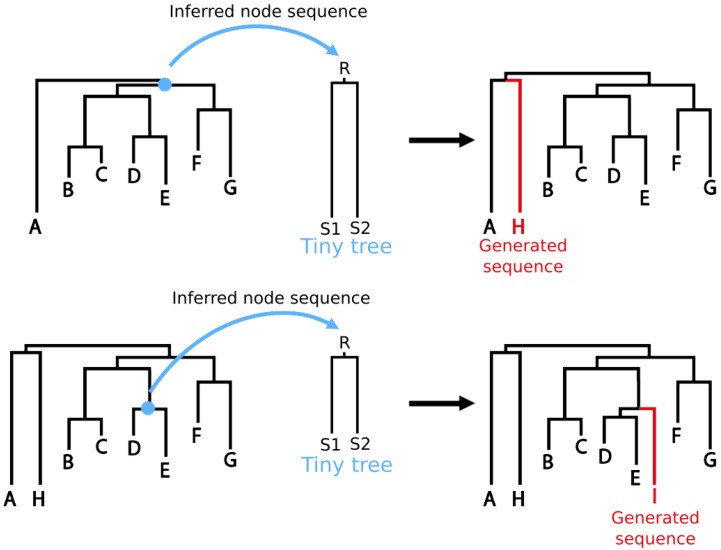
Principle of the Virus Pop simulation process. Starting from the phylogenetic tree inferred from real data, selected internal nodes and their reconstructed ancestral sequences, each simulation is generated with IQ-TREE Alisim tool. The tool is launched on a “tiny tree” made of an ancestral sequence at the root (R), and with two branches which length corresponds to the targeted evolutionary distance. When inferred back in the phylogenetic tree, the new sequences should be inserted at the desired topological position.

**Figure 4 viruses-15-01227-f004:**
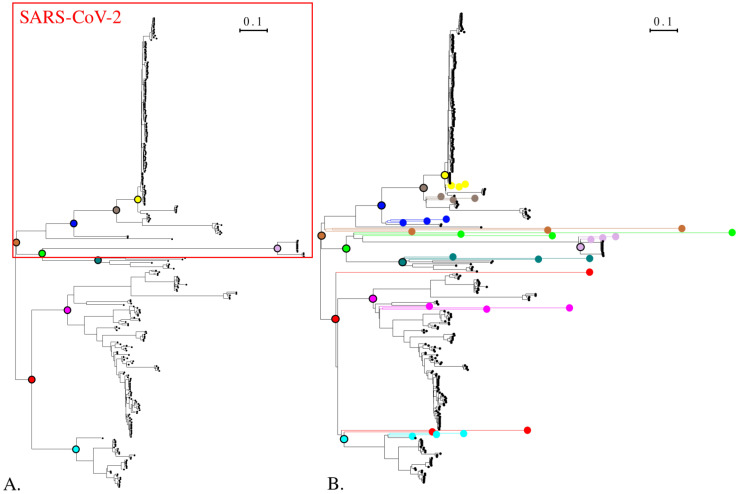
(**A**). The *Sarbecovirus* dataset phylogenetic tree of the spike protein. The tree was built with IQ-TREE. Solid colored circles indicate the internal nodes selected for ancestral sequence reconstruction and simulation. (**B**). Phylogenetic tree of the *Sarbecovirus* spike protein dataset completed with sequences simulated with the Virus Pop pipeline. The model used for the tree constructions is the model inferred by IQ-TREE for the simulation. Solid circles with black outline are the closest equivalent to the selected ancestral nodes in (**A**). Corresponding simulated sequences are solid circles of the same color located at the tree leaves. Three sequences with varying evolutionary distances were simulated from each ancestral sequence. Sequences and Newick tree are available in Appendix A.

**Figure 5 viruses-15-01227-f005:**
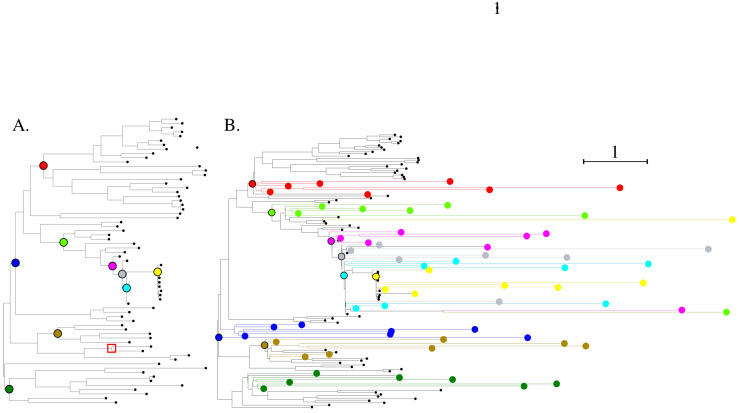
(**A**). The *Circovirus* dataset phylogenetic tree. The tree was built with IQ-TREE. Solid colored circles indicate the internal nodes selected for ancestral sequence reconstruction and simulation. The red square indicates the ancestral node from which we generated the closest sequences to the two new human *circoviruses* studied in Section 3.4. (**B**). Same tree with the simulated sequences. In this example, seven evolutionary distances of 0.1, 0.5, 1, 2, 3, 4 and 5 were simulated from each reconstructed ancestral sequence. The dark green ancestral node is not reported here as the new sequences changed the topology such that there is no equivalent node.

**Figure 6 viruses-15-01227-f006:**
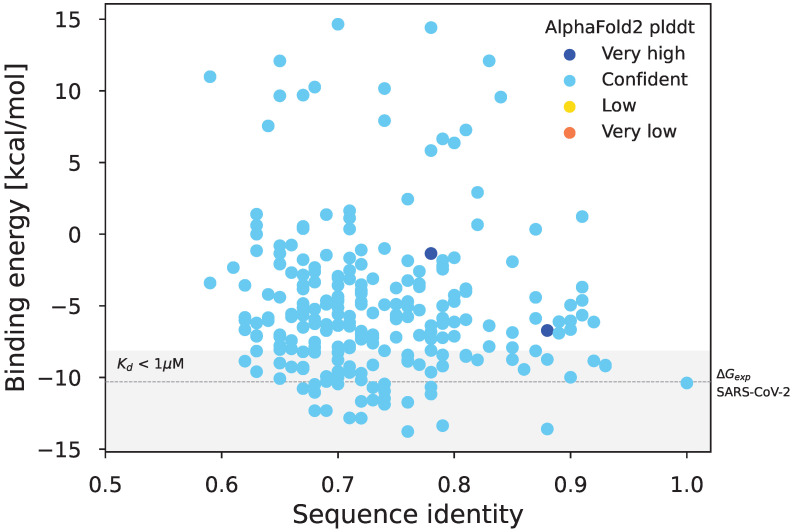
Binding energy vs sequence identity of simulated sequences. The identity is measured by comparing the simulated RBD sequences with the SARS-CoV-2 reference RBD sequence. Colors correspond to ranges of per-residue confidence estimation (plddt) of the structure as constructed by AlphaFold. Very high: >90 and Confident: [70–90]. The gray area corresponds to a binding energy low enough for the complex to be stable (Kd < 1 µM) while the dashed line indicates the experimentally measured binding free energy for the complex formed with the reference SARS-CoV-2 RBD sequence [29].

**Figure 7 viruses-15-01227-f007:**
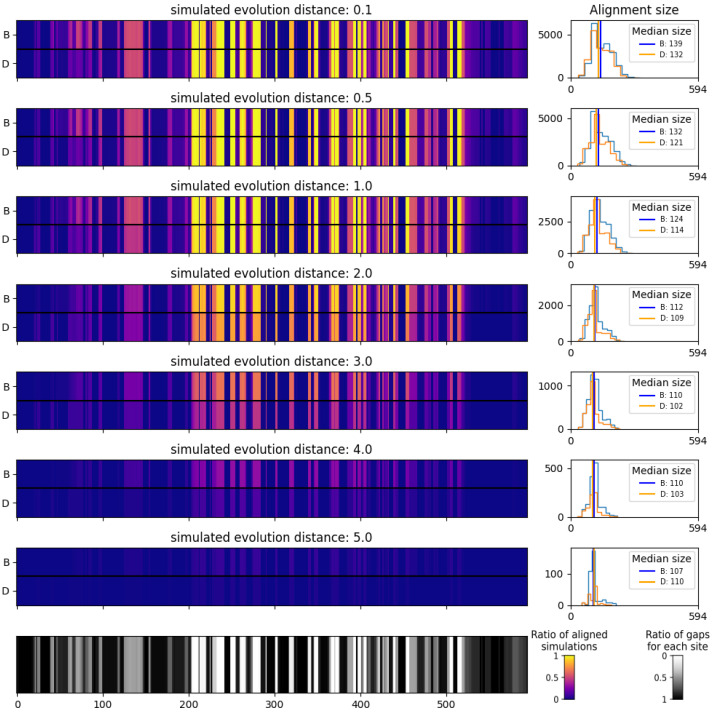
Alignment results of BLASTp and DIAMOND on 6300 simulated *Circovirus* Capsid protein against real *Circovirus* Capsid protein, with varying simulated distances. **Left**: ratio of sequence aligned at each site and ratio of gaps at each site along the multiple sequence alignment. For each alignment plot, BLAST (B) and DIAMOND (D) results are displayed. **Right**: histograms of the alignment size. Only alignment with an e-value below 10 and a bit score above 40 were considered.

**Figure 8 viruses-15-01227-f008:**
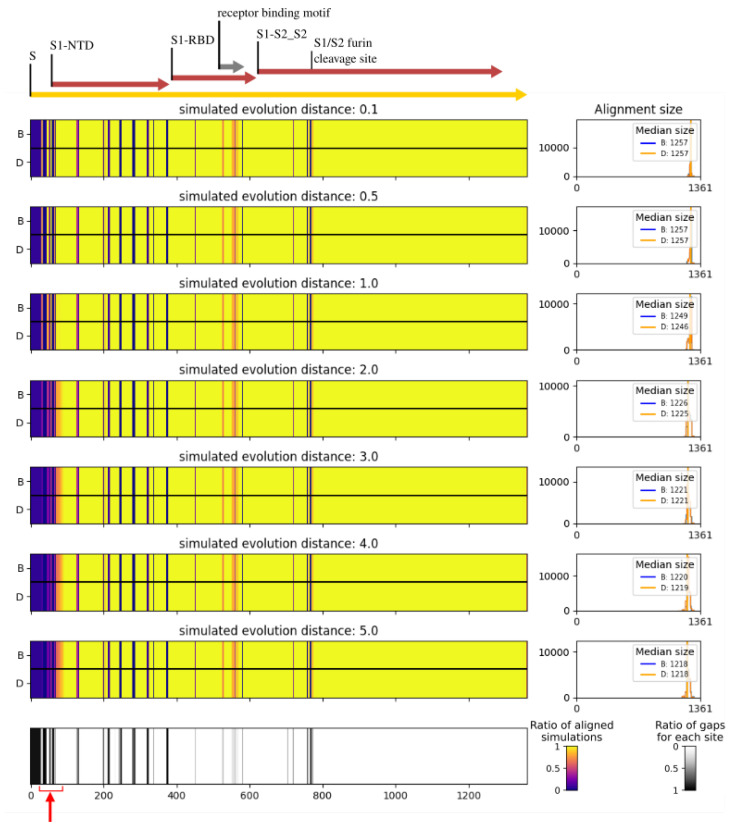
Alignment results of BLASTp and DIAMOND on simulated *Sarbecovirus* spike protein against real spike proteins, with varying simulated distances. Analyzed sequences are available in Appendix A Left: ratio of sequence aligned at each site and ratio of gaps at each site along the multiple sequence alignment. For each alignment plot, BLAST (B) and DIAMOND (D) results are displayed. Right: histograms of the alignment size. Only alignment with an e-value below 10 and a bit score above 40 were considered. The red arrow indicates the only region in which a decrease of the performances is visible. The regions’ limits are based on the reference SARS-CoV-2 sequence.

**Figure 9 viruses-15-01227-f009:**
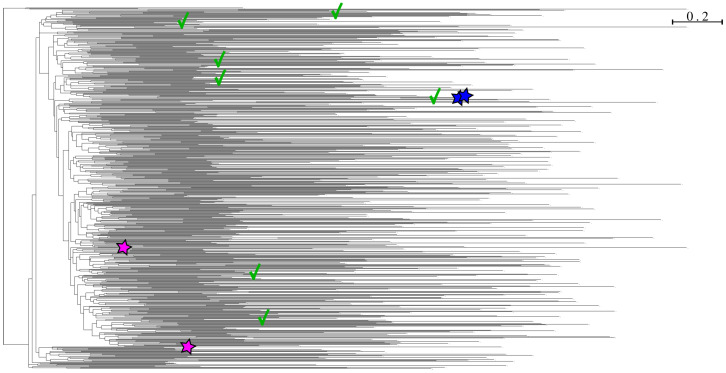
Phylogenetic tree of the two new capsid proteins (blue stars), the two closest sequences in the database (pink stars) (MW686209 and MH603564) and the 600 simulations based on the closest common ancestor of the two closest sequences. This common ancestor is highlighted in Figure 5. The green check marks indicate the simulated sequences on which ten or more reads were aligned. The tree was generated with IQ-TREE.

## Data Availability

The Virus Pop git repository is available at https://gitlab.pasteur.fr/tbigot/viruspop (accessed on 1 May 2023). The Virus Pop database is available at https://doi.org/10.5281/zenodo.7712690.

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
