# Peer review of "Virus Pop—Expanding Viral Databases by Protein Sequence Simulation"

_viruses, 2023, doi:10.3390/v15061227_

Round 1

Reviewer 1 Report (Previous Reviewer 2)

Comments and Suggestions for Authors

Authors have improved the manuscript by responding to all the comments. 

Author Response

We thank the reviewer for the review of the latest version of the manuscript and noticed that he/she found the paper suitable for publication in its current version

Reviewer 2 Report (Previous Reviewer 1)

Comments and Suggestions for Authors

My main concern was about the feasibility of simulating multiple viral family at once, ideally all of them, which was answered by the authors by providing a database with different viral families already simulated. Although I disagree that a tool which works on protein sequences and without NGS data should be published in a special issue entiled "Applications of Next-Generation Sequencing etc.", such decision is due to the editor, so if he/she found that the work had met the expectation of this special issue, for me the paper is ok for publication.

Author Response

We thank the reviewer and noticed that he/she considers that the manuscript is suitable for publication in its current version.

This manuscript is a resubmission of an earlier submission. The following is a list of the peer review reports and author responses from that submission.

Round 1

Reviewer 1 Report

Comments and Suggestions for Authors

Kende et al presented a novel bioinformatics tool which aim is to expand viral databases using simulated sequences in order to improve the ability of metagenomics tools to identify novel viruses too much distant to known ones. The idea is very intriguing and could be a very elegant solution to the problem of identify really unknown viruses for which no related sequence is known.

However, I identified two really big problems with this work. First of all, one problem is related to the special issue of Viruses in which the paper was proposes: "Applications of Next-Generation Sequencing in Virus Discovery 2.0". Despite the title and the abstract partially assessing that the work is related somehow to the NGS (Next Generation Sequence), there is absolutely no result about something that could be used with the NGS. The proposed tool is mainly applied (and based) to the philogeny, which is performed on sequences produced from NGS data or simply from Sanger sequencing. I strongly suggest to send the work to a different special issue or maybe to another journal, but I am sure that in this topic the Editor would make the right decision about.

Second big problem is that results, even if they are promising is demonstrating the potential usefullness of the work, are not really describing a real situation a so a real usage of the proposed tool itself. In case I want to do viral discovery on unknown viruses, by definition I will not know which family my unknown virus belong to, so I will need to add simulated sequences to all knonw viral families and I will need to try to add different number of sequences to each family. Based on the results presented on the perfomances of the tool, I feel that such task will take a high amount of time, so huge to make the usage of the tool itself not really feasible, probably even with the implementation of parallelization, which currently is lacking. So, to sum up, even if the tools is interesting and has potential, I cannot recommend its publication in this special issue nor in Viruses.

Reviewer 2 Report

Comments and Suggestions for Authors
